# Selecting and Performing Service-Learning in a Team-Based Learning Format Fosters Dissonance, Reflective Capacity, Self-Examination, Bias Mitigation, and Compassionate Behavior in Prospective Medical Students

**DOI:** 10.3390/ijerph16203926

**Published:** 2019-10-16

**Authors:** Alexis Horst, Brian D. Schwartz, Jenifer A. Fisher, Nicole Michels, Lon J. Van Winkle

**Affiliations:** Department of Medical Humanities, Rocky Vista University, 8401 S. Chambers Road, Parker, CO 80134, USA; ahorst@rvu.edu (A.H.); bschwartz@rvu.edu (B.D.S.); jfisher@rvu.edu (J.A.F.); nmichels@rvu.edu (N.M.)

**Keywords:** critical reflection, professional development, medical humanities, reflection-on-action, self-appraisal, team-based learning, humanistic values, empathy

## Abstract

More compassionate behavior should make both patients and their providers happier and healthier. Consequently, work to increase this behavior ought to be a major component of premedical and medical education. Interactions between doctors and patients are often less than fully compassionate owing to implicit biases against patients. Such biases adversely affect treatment, adherence, and health outcomes. For these reasons, we studied whether selecting and performing service-learning projects by teams of prospective medical students prompts them to write reflections exhibiting dissonance, self-examination, bias mitigation, dissonance reconciliation, and compassionate behavior. Not only did these students report changes in their behavior to become more compassionate, but their reflective capacity also grew in association with selecting and performing team service-learning projects. Components of reflective capacity, such as reflection-on-action and self-appraisal, correlated strongly with cognitive empathy (a component of compassion) in these students. Our results are, however, difficult to generalize to other universities and other preprofessional and professional healthcare programs. Hence, we encourage others to test further our hypothesis that provocative experiences foster frequent self-examination and more compassionate behavior by preprofessional and professional healthcare students, especially when teams of students are free to make their own meaning of, and build trust and psychological safety in, shared experiences.

## 1. Introduction

“If you want others to be happy, practice compassion. If you want to be happy, practice compassion.”—Dalai Lama

More compassionate behavior should make both patients and their providers happier and healthier [1,2]. Consequently, work to increase this behavior ought to be a major component of premedical, medical, and healthcare professions education. Compassionate professional behavior is believed to be “animated” by humanistic values such as altruism, duty, excellence, honor and integrity, accountability, and respect for others [3,4]. These abstract values are difficult for students to apply, however, without concrete experiences [5]. Hence, we predicted that the experiences of selecting and performing service-learning projects would help teams of prospective medical students enhance their compassionate behavior. These projects were expected to arouse dissonance and its reconciliation through critical reflection (CR). In the definition of CR we provided to students—one recognizes how their thoughts and behaviors do not match their personal and humanistic values, experiences perplexity, doubt, hesitation, or mental difficulties (i.e., dissonance), and decides how better to align their values, thoughts, and behaviors (i.e., dissonance reconciliation) [6,7,8]. For purposes of the present study, we concluded that students used self-examination when they gained insight into how better to align their thoughts and values. We defined CR as aligning both thoughts and behaviors with personal and humanistic values.

For example, in the following reflection, published by Hernandez et al. [9], the student exhibited both self-examination and compassionate behavior (CR).
Part of my application to medical school was a story about my grandfather…He’s the person I probably most respect in this world, and he doesn’t speak English, and I remember going with him as a kid to the doctor…it’s one of my driving factors in medicine, and I’m always willing to interpret [for patients]. I’m always willing to stay behind. I never complain…because I know…people deserve that, and…people probably have grandchildren in medical school or who are doctors, so I pray that [healthcare professionals] really treat…people the best they can.

Had this reflection stopped where the student realized he/she should interpret for others (not just his grandfather) but had not yet done so, the quote would qualify as self-examination (according to our definition). The full quote includes compassionate professional behavior (CR) because the student voluntarily interprets for other patients, apparently whenever possible, because of insight he/she gained while interpreting for her/his grandfather.

### 1.1. The Importance of Dissonance, Implicit Bias, and Their Reconciliation Using Self-Examination and CR

While cognitive dissonance often leads to self-deceptive rationalization to resolve the discordance, it can also produce the perplexity, doubt, hesitation, or mental difficulty needed to help people better align their values, thoughts, and behaviors. That is, to seek authenticity through virtue ethics. For example, Barkan and associates [10] reviewed much of the literature concerning ethical dissonance and described how and why people fail to live up to their ethical values by, say, “shuffling and stretching the truth.” They also conclude, however, that “ethical dissonance can harness the tension it creates to help people uphold their aspired moral standards.”

In an educational setting, explicit discussions about implicit bias have been used in elective workshops for third-year medical students to produce dissonance and two types of reflection to reconcile vs. rationalize the dissonance [9]. Bias mitigation and more compassionate behavior are important because implicit biases against patients adversely affect their treatment, adherence, and health outcomes, apparently owing to low compassion in interactions with their providers [11]. In about half of their written reflections, the students of Hernandez and associates [9] referred to personal values to reconcile their dissonance and mitigate their now more conscious biases. Students, who reconciled dissonance, experienced professional development and positive changes in their attitudes. In the other half of the reflections, however, students used normative standards to preserve their existing beliefs about their thoughts and behaviors. As a result of the latter rationalization, students exhibited no self-examination, CR, or development and retained their original positions [9].

In contrast to the approach of Hernandez et al., we did not raise the issue that prospective medical students might hold implicit biases against some groups of people. Instead, we provided teams of prospective medical students with required opportunities to experience dissonance and negative biases, simply through selecting and performing service-learning projects for a medical humanities course (e.g., to identify a nursing home at which they subsequently volunteered). We did not explicitly mention biases, nor did we intentionally discuss biases in our course. We merely suggested to students that they pay attention to the dissonance they may experience in association with their projects, as defined above for CR. Through these vivid and current experiences, we expected to foster self-examination and performance of explicitly more compassionate behavior as described by students in their written individual and team reflections.

### 1.2. Team-Based Learning (TBL) Produces a Welcoming Team Environment for Reflection

Michaelsen and associates [12] describe TBL as transformative. According to these authors, it “transforms ‘small groups’ into ‘teams’, makes a technique a strategy, increases the depth and quality of student learning, and (for many teachers) restores the joy of teaching.” TBL is a teaching strategy distinct from most small group learning methods. Its purpose is to produce learning teams that exceed the creativity and productivity of any team member. Consequently, every individual team member learns more. In general, TBL can be distinguished from other types of small group learning in that teams are characterized by a high level of individual commitment to the welfare of the team and a high level of trust among team members [12,13]. Thus, TBL can provide safe spaces for self-examination and CR by students in teams [8,14,15,16,17].

### 1.3. Hypotheses

These considerations led us to formulate four hypotheses:

**Hypothesis 1**:
*The direct and vivid experiences of selecting and performing service-learning projects by teams of prospective medical students in a medical humanities course will be associated with written reflections exhibiting dissonance, self-examination, bias mitigation, dissonance reconciliation, and compassionate behavior.*


**Hypothesis 2**:
*Students will express positive attitudes toward their teams, team-based learning, community service, and recognizing/mitigating their implicit biases in association with team service-learning experiences.*


**Hypothesis 3**:
*Reflections on team service-learning experiences will be accompanied by increases in students’ reflective capacity, as measured by a reliable survey of this characteristic.*


**Hypothesis 4**:
*Students’ reflective capacity scores will correlate positively with their cognitive empathy scores (a component of compassion [18]).*


## 2. Methods

### 2.1. Participants

Three cohorts, of twenty-six prospective medical students each, enrolled and were studied in a medical humanities course from August through December 2017 (first cohort), 2018 (second cohort), or 2019 (third cohort in progress). The course was part of a Master of Science in Biomedical Sciences (MSBS) program for prospective medical students at Rocky Vista University in Parker, Colorado, US. The main objective of the course was to provide teams of MSBS students opportunities to form and develop their personal and professional skills and identities through readings, quizzes, classroom application discussions, and written reflections on service to the community. Complementary criteria were used to assess development of students in each cohort.

### 2.2. Team Formation and Procedure

Teams of six or seven students were formed on the first day of class. Students’ gender identities were distributed as equally as possible across the 4 teams in each cohort. These teams not only performed service-learning projects, but they also worked together to take team quizzes in a team-based learning format [12,15].

Teams were encouraged to select service-learning projects that related to some part of the medical humanities course content (e.g., meeting and communicating with residents of a nursing home). They sought approval for their projects from the course director, but otherwise selected and performed their projects on their own. The work was divided among team members such that each team member performed at least four of the 28 h of service that were required of each team. Teams met initially to select and plan their projects and then regularly at 4- to 5-week intervals to discuss the progress of their projects. These team meetings were scheduled by teams at their convenience, and the meetings were used to discuss the written reports and reflections of students planning or performing the work of the project at their community service sites over the preceding 4 to 5 weeks.

Teams generated written minutes and reflections from each meeting. In these minutes and reflections, teams integrated their service-learning-related experiences with the humanities course content. For example, students studied communication in our course, so they often referred to effective and ineffective communication with people they were serving. No specific prompts were provided to students or teams, however, in order to allow them to decide how best to reflect on their experiences.

Each student produced more than one page of reflections for every 2 h of community service, and teams usually generated at least one page of minutes and reflections for each team meeting. Together, the minutes and reflections accounted for 52% of the grades assigned to students in the one credit-hour course. These minutes and individual and team reflections were submitted to the course director no later than the ends of the 2nd, 6th, 11th, and 16th week of the 17-week semester.

### 2.3. Assessment of Dissonance and Subsequent Development by Students in the First Cohort

To generate the data presented here, two investigators (AH and LV) independently assessed discussion minutes and reflections for the presence of dissonance, self-examination, CR, dissonance reconciliation, preservation, and bias toward other people or venues, as defined previously [7,14,17]. According to our definition, self-examination (including CR) was exhibited when a student recognized how their thoughts (and behaviors) did not match their personal and humanistic values; experienced perplexity, doubt, hesitation, or mental difficulties (i.e., dissonance); and began to decide how better to align their thoughts (and behaviors) with their values (i.e., dissonance reconciliation) [6,7,8]. That is, self-examination (and CR) were present only when dissonance and dissonance reconciliation both occurred. This definition is consistent with the model of reflection proposed by Nguyen and associates [19]. See Appendix A and Appendix B for examples of dissonance, self-examination, dissonance reconciliation, bias mitigation, and (in the case of Appendix B) compassionate behavior and CR. For readers interested in further discussion, full sets of anonymous individual and team reflections are available from the corresponding author on request.

For the purpose of grading, the qualitative assessments for self-examination (and CR) by the course director (LV) were assigned scores ranging from 80% (thinking and reflection but no self-examination exhibited) to 100% (self-examination exhibited). Partial self-examination (e.g., dissonance only) usually was assigned a score of about 90%. Scores assigned by independent assessors for self-examination correlate well, as we reported elsewhere (*r* = 0.92, [17]). We refrained from judging the content of written reflections except for the presence of dissonance, bias, self-examination, CR, dissonance reconciliation, and preservation.

Students also completed an in-house survey of their attitudes toward team service-learning projects (Table 1) after they finished them in December 2017. Completion of the survey by students was optional and anonymous. The response rate was 100%.

### 2.4. Measurement of Development by Students in the Second Cohort

Due to the fact that we wanted to measure whether reflection on service-learning correlated with an increase in students’ reflective capacity, we invited the second cohort of prospective medical students to complete the 40-item Reflective Practice Questionnaire (RPQ) [20,21] prior to an introduction to their medical humanities course in August 2018 and after completion of their team service-learning projects in December 2018. The RPQ is a reliable measure of reflective capacity and related characteristics—desire for improvement, general confidence, confidence communicating with patients/clients, uncertainty, stress interacting with patients/clients, and job satisfaction (see Rogers et al. [21] online for the current version of the RPQ). Students in the second cohort also completed the in-house survey of their attitudes toward team-based learning, community service, and implicit biases. Completion of the surveys was optional and anonymous. The response rates for both surveys were 100%.

### 2.5. Use of the Third Cohort to Determine Whether Reflective Capacity Correlates with Cognitive Empathy

Students in the third cohort completed the RPQ and the Jefferson Scale of Empathy (JSE, HPS-Version), a valid and reliable measure of cognitive empathy [18], just prior to an introduction to the medical humanities course on the first day of class. Students marked their surveys with an ID code so each student’s RPQ scores could be matched to their JSE score. Completion of the surveys was optional and anonymous. The response rates for both surveys were 100%.

### 2.6. Statistical Analysis

Statistical analyses were performed using GraphPad Prism 8.0.2 Software, Inc. (La Jolla, CA, USA). Since much of our data were not normally distributed (e.g., Table 1), we report some results as medians rather than means and used nonparametric statistics to analyze them. Whether students’ median survey opinions—about their team service-learning projects—differed significantly from neutral were determined using one-sample Wilcoxon tests, and the medians of these opinions were compared statistically using the Kruskal–Wallis statistic. Similarly, the median frequencies at which students in the first cohort expressed dissonance, self-examination, dissonance reconciliation, bias mitigation, compassionate behavior (CR), and preservation (out of four opportunities to do so) were compared statistically using the Kruskal–Wallis statistic in association with nonparametric multiple comparison tests to compare the frequencies of the different dependent variables. We also used one-sample Wilcoxon tests to determine whether the latter medians differed significantly from zero. Grubbs’ test to detect an outlier was used to determine whether any student’s value was a statistically significant outlier in each set of data.

For the second cohort, students’ initial mean reflective capacity (RC) score in August was compared to their final mean score in December using an unpaired *t*-test and ANOVA with multiple comparison tests (when these mean RC scores were compared to a prior mean score of medical students). The latter *t* and F values were used to calculate effect size (ES) as an *r* value using the GraphPad Prism 8.0.2 Software [22]. Correlations between and among RC and other characteristics measured by the RPQ were determined as Pearson *r* values using the same software. In the third cohort, correlations between RPQ and JSE scores were also determined as Pearson *r* values.

This study was reviewed and found to fulfill the criteria for exemption by the Rocky Vista University Institutional Review Board (IRB). Written informed consent was obtained from all 26 students in the first cohort to analyze, report, and publish their written reflections for the medical humanities course. See summary of studies in Table 2.

## 3. Results

Our data support each of our four hypotheses well.

**Hypothesis** **1:**
*The direct and vivid experiences of selecting and performing service-learning projects by teams of prospective medical students in a medical humanities course will be associated with written reflections exhibiting dissonance, self-examination, bias mitigation, dissonance reconciliation, and compassionate behavior.*


### 3.1. Dissonance and Bias in the First Cohort

In support of hypothesis 1, most students experienced dissonance in their four opportunities to reflect (i.e., they recognized that their thoughts or behaviors did not match their personal and humanistic values, and they experienced perplexity, doubt, hesitation, or mental difficulties). Most of them then used self-examination to reconcile the differences between their thoughts or behaviors and humanistic values. Implicit biases toward people or settings caused about half of the incidences of dissonance (Figure 1).

For example (from Appendix A);
My last reflection, I investigated my anxiety for our volunteer project selection. How, I wanted to work in the hospital compared to the homeless shelter. I wrote about some reasons for these emotions but during this week’s group meeting, I think I pin-pointed where my anxiety stems from…
I have noticed that just walking sixteenth street mall gives me the same feelings that I first experienced that day. I feel scared and intimidated by the thought of even walking by homeless [people]. No matter what color, I have even noticed that I try to cross the street before making any sort of eye contact with them.
I know that my experiences were traumatic, however I cannot let that one experience completely change my view of the entire homeless population…

According to both assessors (AH and LV), all but one student moved from reflections on the experiences they anticipated during service-learning, to CR (including compassionate behavior) during or after service to the community. The median frequency of this CR was one time out of two opportunities to do so in the third and fourth reflections (Figure 1). In line with our definitions in the Introduction, this self-examination and CR included reports in written reflections of compassionate behavior more consistent with students’ humanistic values. As illustrated in Appendix B;
…Panic, fear, and dread overwhelmed me, and I could not control it. So, instead I embraced it. For the entirety of the drive I simply did not fight the thought but instead tried to feel them out and rationalize how ‘off the wall’ they were and made a pact with myself that after this drive these feelings will be let go. And, that is what I did.
The moment I stepped out of the car, I felt anew. My feelings of dread turned to excitement. An ease came over my mind that is hard to put into words. However, walking in the door, (I know this sounds cliché) but a new man. As part of my job, I stood by the front door and greeted every person that walked through the door that day. It was a true joy, as I no longer felt any fear or anxiety that was gripping me in the car as every person I interacted with during my hours treated me with the utmost respect…

### 3.2. Value of Students’ Unique Team Experiences

There were some differences in the performances among teams of students. For example, members of one team exhibited self-examination somewhat less frequently than the other three teams (Figure 2a). These students failed, at first, to focus on their dissonance, so self-examination apparently seemed less warranted to them. The team sought help from the course director, however, and improved their frequency of self-examination significantly between the first two and the final two opportunities for written reflection (two-by-two contingency table, X^2^ = 7.34, ES = *r* = 0.51, GraphPad Prism 8.0.2 Software, *p* < 0.01) [22].

In providing feedback to teams concerning their written reflections, we used the “yes and” approach of improvisation [23]. Using this approach, we focused on what a team was doing well and how they could expand their efforts to accomplish reflection that was more introspective and critical in nature. We did not seek simply to “soften the blow” of less than perfect grades. Rather, we enlisted team members in the effort to move them to their best performances [23]. No other guidance or prompts were given to teams of students in order to give them the maximum possible flexibility in creating their own meaning, minutes, thoughts, and reflections [24].

There was also a large difference in the source of dissonance among teams. According to both assessors, members of one team reconciled dissonance resulting from bias much more frequently than the other three teams (Figure 2b). Interestingly, implicit bias against homeless people was expressed most frequently in response to the final question in Table 1 “Of what biases did you become aware during encounters with people/venues in your service-learning project?” (Table 3), and the team that worked with homeless people was the team that reconciled bias most often (Figure 2b).

Support from their team members also likely contributed to development of students’ reflective capacities. Conversely, self-examination in teams likely led students to further value their teams. Recall the example above:
*…during this week’s group meeting, I think I pin-pointed where my anxiety stems from…* (Appendix A).

**Hypothesis 2**:
*Students will express positive attitudes toward their teams, team-based learning, community service, and recognizing/mitigating their implicit biases in association with team service-learning experiences.*


In the survey concerning team- and service-learning, students overwhelmingly expressed attitudes supporting hypothesis 2. All but one student disagreed, and 62% disagreed strongly, with the statement “I would have been better off on another team in medical humanities”, and most of them agreed that “All things considered, I could not have been assigned to a stronger team in medical humanities” (Items 2 and 4 in Table 1). Consequently, all but one student agreed that “Medical Humanities should continue to use team-based learning in future courses,” and no student disagreed with this statement (Item 6 in Table 1).

Similarly, 25 of the 26 students agreed with the statements “Encounters with people/venues in our service-learning project helped me to see my potential biases toward people/venues more clearly”, “Writing reflections on our service-learning project fostered my professional development”, and “Encounters with people in our service-learning project will help me to be engaged with people regardless of the setting or disposition of the person” (Items 7, 9, and 10 in Table 1).

One student in the first cohort was, however, a statistically significant outlier in response to the latter survey (Table 1), and the contents of one student’s written reflections were also statistically significant outliers (Figure 1). The latter student appeared to experience no dissonance and so did not use self-examination or CR to reconcile it. Similarly, the former student felt strongly that service-learning and written reflections were of little value to her/him (Items 1, 3, 5, 7, 9, and 10 in Table 1). Based on these results, the former and latter student(s) may be the same person. In contrast, virtually all other students valued service-learning and written reflections about their experiences (Items 1, 3, 5, 7, 9, and 10 in Table 1). The responses of students in the second cohort were virtually the same as those shown for the first cohort in Table 1 (data not shown for the second cohort).

**Hypothesis 3**:
*Reflections on team service-learning experiences will be accompanied by increases in students’ reflective capacity as measured by a reliable survey of this characteristic [20,21].*


Consistent with hypothesis 3, reflective capacity (RC) increased in prospective medical students in the second cohort in association with our medical humanities course (Figure 3, *r* = 0.28, *p* < 0.05). Also consistent with hypothesis 1, more compassionate behavior (CR) likely occurred in association with reflection on team service-learning by the second cohort. Self-appraisal and reflection-on-action are components of the RC survey [20,21], and both self-appraisal (r = 0.27, *p* < 0.05) and reflection-on-action (r = 0.29, *p* < 0.05) contributed to the increase in RC observed in prospective medical students.

Similarly, other measures of the RPQ were consistent with positive and healthy developmental changes in prospective medical students between August and December 2018. In August, uncertainty and stress interacting with patients/clients were each negatively associated with both general confidence and confidence communicating with patients/clients (Table 4). By December, these statistically significant negative associations were lost, and uncertainty and stress were instead each positively associated, in a statistically significant manner, with desire for improvement (Table 5). In association with these changes, both uncertainty and stress went from negative to positive relationships with reflective capacity (RC) (Table 4 and Table 5, Fisher r-to-z transformation, *p* = 0.05 for the change in the Pearson correlation values of uncertainty with RC and stress with RC).

**Hypothesis 4**:
*Students’ reflective capacity scores will correlate positively with their cognitive empathy scores (a component of compassion [18]).*


For students in the third cohort, RC, self-appraisal, and reflection-on-action scores each correlated strongly with JSE scores (*r* = 0.72, 0.72, and 0.69, respectively, *p* < 0.0001).

## 4. Discussion

### 4.1. Impact of Dissonance in the First Cohort

All but one student in the first cohort experienced dissonance in association with selecting and performing a service-learning project. This dissonance was the focus of their written CR in the 3rd and (or) 4th reflections to reconcile their thoughts and behaviors with their personal and humanistic values. Prior to CR describing improved behavior in association with service to the community, most students reflected about the behavior they anticipated during service (i.e., in their 1st and/or 2nd written reflections).

In about half of the cases, this work to reconcile dissonance concerned bias against people or venues (Figure 1, Table 1 and Table 3). In this regard, most healthcare professionals in the US exhibit positive implicit bias toward white people and negative attitudes toward people of color [11,25]. Such biases appear to influence patients’ treatment, adherence, and health outcomes, apparently owing to low compassion from their providers. Furthermore, bias in residency selection could deprive patients of care by the most qualified healthcare providers [26,27]. Hence, methods such as ours to mitigate bias in virtually every prospective medical student could help improve public health in general and the health of people of color in particular. To do so, however, likely requires changes in the culture of medicine, such that efforts to mitigate bias through reflection on service to the community continue throughout providers’ careers.

Bias mitigation by all but one of our students is consistent with the model of Thompson and associates in which such dissonance can lead to reconciliation through self-examination and CR [7]. In the latter authors’ elective workshops for third-year medical students, however, students reconciled dissonance, experienced desirable professional development, and changed their attitudes only about half of the time as a result of explicit discussions about implicit bias [9]. In the other half of their written reflections, these students preserved existing beliefs about their thoughts and behaviors. Perhaps explicit assignments and discussions about implicit bias elicited defensive reactions by students in half of the reflections.

However, self-examination and implicit bias mitigation can be performed without teaching or even suggesting to students that they might hold such biases. Hypothetically, all students need are their own experiences and the freedom to respond to them. We used the experiences of selecting and performing team service-learning projects to foster dissonance on which to reflect (Figure 1). No other requirements, instructions, or prompts were given to students in order for them to have freedom to identify and focus on their dissonance and work to reconcile it.

Dissonance reconciliation is also valuable for busy healthcare professionals who might otherwise be inclined to rationalize less than ideal behavior. For example, nursing care erosion in hospitals has occurred because of dissonance followed by rationalization and preservation instead of self-examination and CR [28], such as when a nurse believes he should explore patients’ emotional states more thoroughly than he does but rationalizes that he does not have enough time to do so. Such dissonance could, however, be reconciled positively through self-examination and CR to help nurses and other healthcare professionals better align their thoughts and behaviors with their humanistic values and ideals [29].

Helping people become more aware of such psychological struggles can move them to reconcile their dissonance through self-examination and CR rather than rationalization and preservation [30]. We helped to make our students aware of their struggles simply through the assigned work, course structure, and provision of the dissonance-focused definition—critical reflection is exhibited when one recognizes how their thoughts and behaviors do not match their personal and humanistic values; experiences perplexity, doubt, hesitation, or mental difficulties; and begins to decide how better to align their values, thoughts, and behaviors.

In the present study, the dissonance resulting from selecting and performing service-learning fostered professional development through self-examination, bias mitigation, and self-reported, compassionate behavior (i.e., CR) in virtually all prospective medical students (Figure 1). In our view, self-examination and CR are often achieved with concurrent, thought-provoking experiences. Supporting our current findings, we found previously that students performed self-examination much more frequently when they elected to reflect on concomitant experiences rather than the topics for reflection we assigned [14,16,17]. Students were given flexibility to create their own meaning in their written thoughts and reflections [24]. They did not receive prompts or a detailed rubric of expectations for their written reflections. Reflections were assessed only for the presence of self-examination, CR, and related characteristics described above. (See Methods.)

### 4.2. Team Support Also Likely Fostered Self-Examination and CR

Despite their importance, personal disclosures were not required of students in our course so that they would not feel overexposed. They received acceptable assessments even without such disclosure. Nevertheless, work in teams seemed to help make personal disclosure safer and more comfortable as teams developed. Moreover, students likely felt less exposed in their written reflections since teams, not individual students, received assessments for written reflections [8,14,15,16,17]. In these ways, we believe we were able to assuage the vulnerability inherent in reflective writing and writing in general.

We also believe we fostered self-examination and CR by having students work in teams to select and perform service-learning projects and write reflections. They valued their teams, and clearly benefited from team support (Items 2, 4, and 6 in Table 1). Teamwork fosters trust and psychological safety [31], which helps students face the vulnerability inherent in writing in general and self-examination and CR in particular. Our prospective medical students built trust and safety in their teams through team-based learning in class as well as team meetings, outside of class, to discuss and reflect on their service-learning projects.

### 4.3. How Do Results with the Second Cohort Complement and Support Conclusions Regarding the First?

Reflections with others, performed by teams in both the first and second cohorts, are a valuable component of reflective capacity (RC), which we hoped to foster through team-based learning (e.g., Appendix A). Initially, students in the second cohort had a relatively high mean reflection with others score, which was statistically indistinguishable from that of mental health practitioners in a prior study [20]. Consistent with our hope to foster reflection with others, our students’ mean reflection with others score increased to exceed that of mental health practitioners by the end of our medical humanities course (ANOVA, *r* = 0.28, *p* = 0.01).

RC also increased in prospective medical students in association with our medical humanities course (Figure 3). This increase in RC supports the theory that reflection on service-learning fosters compassionate behavior in students (e.g., Appendix B). Both reflection-on-action and self-appraisal are components of the RPQ, and both contributed significantly to the increase in RC of students in the second cohort. Presumably, reflection-on-action and self-appraisal contributed to quests for the more compassionate behavior reported by students in our first cohort (Figure 1).

Similarly, students’ uncertainty and stress, that in August related to lower confidence in the second cohort (Table 4), apparently caused them to desire to improve by December (Table 5). Changes in the relationships between RC and uncertainty and between RC and stress, from negative to positive between August and December, also likely contributed to students’ personal and professional development (Table 4 and Table 5, Fisher r-to-z transformation, *p* = 0.05 for the change in each Pearson correlation value). Although we did not assess written reflections by students in the second cohort for reports of more compassionate behavior, such behavioral changes seem likely to have occurred. All but one student in the first cohort reported their own more compassionate behavior in at least one of their final two written reflections, and our data indicate that the attitudes and behavioral changes of the first two cohorts were virtually the same (Figure 1 and Figure 3; Table 1 and similar data not shown for cohort two). Moreover, RC, self-appraisal, and reflection-on action all grew in the second cohort, and each of these characteristics correlated strongly with cognitive empathy (a component of compassion [18]) in the third cohort.

## 5. Limitations

We studied three cohorts of 26 prospective medical students in a medical humanities course at a single university, so our results are difficult to generalize to other universities and other preprofessional and professional healthcare programs. Moreover, we did not study in detail how students’ personalities and past as well as present experiences may have influenced our results. Nevertheless, we reported similar results in biochemistry courses for medical, pharmacy, and prospective medical and dental students at another university [8,14,16,17]. We encourage others to test further our hypothesis that provocative experiences foster frequent self-examination, including CR, by preprofessional and professional healthcare students, especially when teams of students are free to make their own meaning of, and build trust and psychological safety in, shared experiences.

## 6. Conclusions

The experiences of selecting and performing team service-learning projects produced dissonance among virtually all prospective medical students in the first cohort, in part, owing to our instruction to them to try to be aware of such feelings. While dissonance took several forms, increased awareness of biases toward other people produced the dissonance about half of the time. Dissonance led to self-examination and CR instead of rationalization to reconcile the dissonance in 25 of 26 students. Thus, assigning students to reflect in relation to current and provocative individual and team experiences fostered frequent self-examination and, later, CR and improved behavior during service to the community (Table 6).

In a subsequent cohort of 26 prospective medical students, reflective capacity (RC) in general, and reflection-on-action and self-appraisal in particular, grew in association with our medical humanities course according to a reliable survey of RC. This evidence of increased self-examination occurred in association with apparent refocusing of uncertainty and stress on a desire to improve. These findings, using more objective measures of RC, complement and support the conclusion that compassionate behavior, including CR, likely grew in both the first and second cohorts of prospective medical students. RC, self-appraisal, and reflection-on action each correlated strongly with cognitive empathy (a component of compassion) in the third cohort (Table 6). All three cohorts likely benefited from students’ support and trust of each other in teams and the psychological safety associated with team-based learning.

## Figures and Tables

**Figure 1 ijerph-16-03926-f001:**
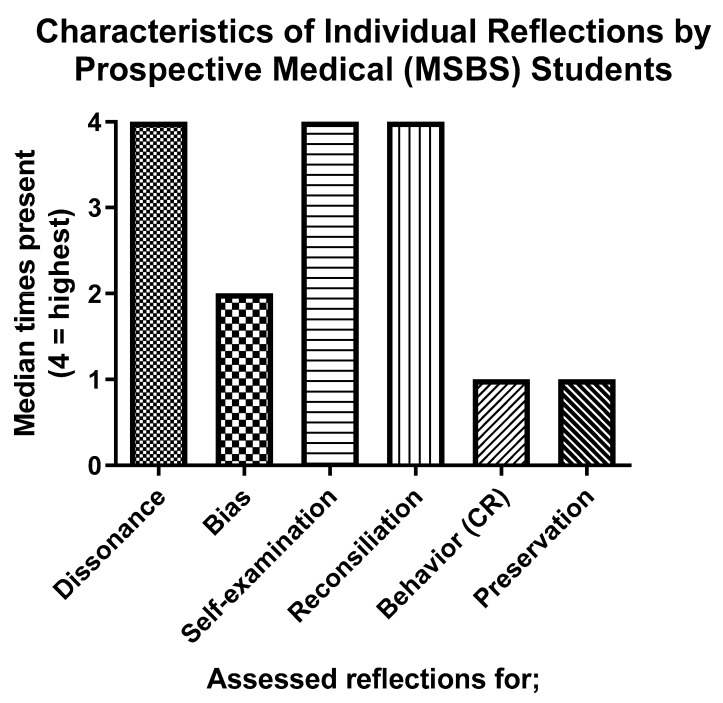
Characteristics of individual written reflections by prospective medical (MSBS) students. The median frequencies of dissonance, bias (and its mitigation), self-examination, dissonance reconciliation, compassionate behavior (CR), and preservation by each student on four separate occasions were calculated from 100 individual written reflections (four per student). One of the 26 MSBS students did not express dissonance on any occasion, and was, thus, a statistically significant outlier (*p* < 0.01) and was not included in the analyses. Dissonance, self-examination, and dissonance reconciliation all occurred more frequently than bias mitigation, compassionate behavior (CR), and preservation (*p* < 0.01). Nevertheless, the medians are all greater than zero (*p* < 0.0001).

**Figure 2 ijerph-16-03926-f002:**
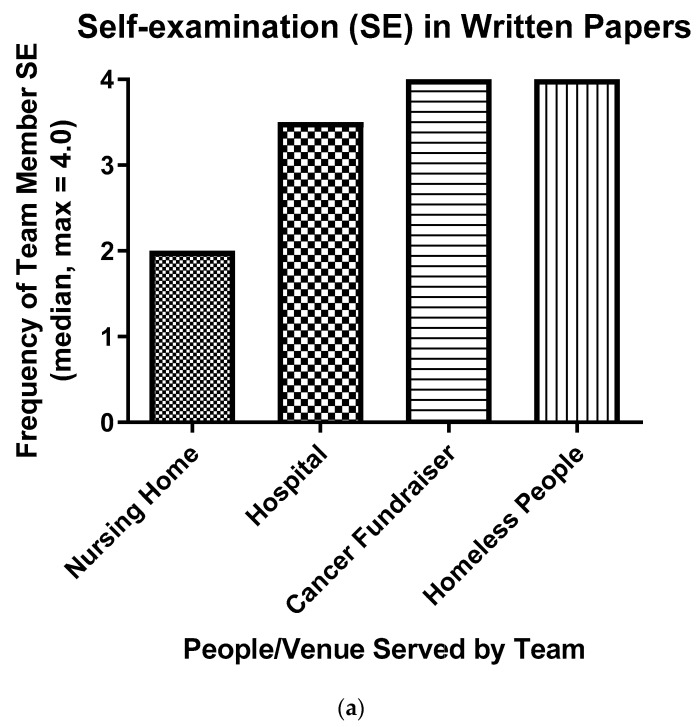
One team performed self-examination somewhat less frequently (**a**) while another team mitigated biases more frequently (**b**) than the other three teams. The median frequencies of self-examination and bias mitigation by each student on four separate occasions were calculated from 100 individual written reflections (four per student and six or seven students per team). One student’s performance was a statistically significant outlier in ‘a’ and another in ‘b’, and they were not included in the analyses (*p* < 0.05). While self-examination occurred somewhat less frequently in one team (*p* < 0.05), and bias mitigation was more frequent in another (*p* = 0.02) (Kruskal–Wallis statistic), each median is significantly greater than zero (*p* < 0.05) (Wilcoxon test).

**Figure 3 ijerph-16-03926-f003:**
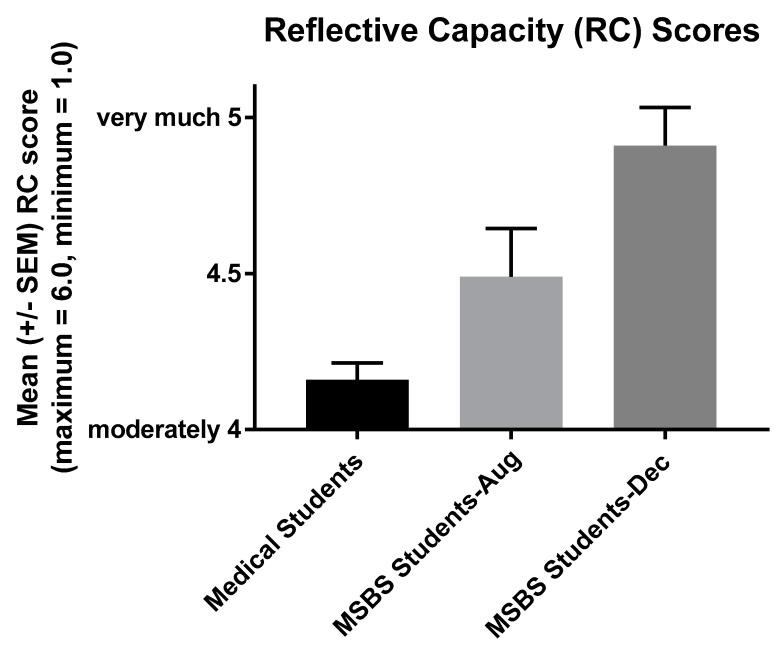
Increase in reflective capacity scores of prospective medical (MSBS) students in association with written reflection on service-learning between August and December 2018. The mean response of MSBS students increased from closer to “moderately” in August to near “very much” in December for 16 statements concerning self-reported reflection. ANOVA with multiple comparison tests revealed both a statistically significant increase in MSBS student scores (*p* = 0.03) and that MSBS student scores in August (*p* = 0.03) and December (*p* < 0.0001) were higher than a sample of graduating medical students [21]. F =16.83, *r* = 0.43, *p* < 0.0001.

**Table 1 ijerph-16-03926-t001:** Median responses to questions regarding team- and service-learning. Except for item 8, each median is different from 4.00 (*p* < 0.0001). Also shown below each question is the distribution of responses.

	1 Strongly Disagree	2 Disagree	3 Somewhat Disagree	4 Neither Agree/Disagree	5 Somewhat Agree	6 Agree	7 Strongly Agree	Median *
1.	Having a team service-learning project in Medical Humanities was very engaging.							6.0
	1 (3.8%) *	0	0	2 (7.7%)	5 (19.2%)	9 (34.6%)	9 (34.6%)	
2.	I would have been better off on another team in Medical Humanities.							1.0
	16 (61.5%)	7 (26.9%)	2 (7.7%)	0	1 (3.8%) *	0	0	
3.	Next year, Medical Humanities should continue to expect teams of MSBS students to perform service-learning projects and to write reflections on their experiences with the projects.							6.0
	1 (3.8%) *	0	0	1 (3.8%)	2 (7.7%)	10 (38.5%)	12 (46.2%)	
4.	All things considered, I could not have been assigned to a stronger team in Medical Humanities.							6.0
	0	1 (3.8%)	2 (7.7%)	1 (3.8%)	3 (11.5%)	9 (34.6%)	10 (38.5%)	
5.	I gained very little from our service-learning project and written reflections on the project.							2.0
	12 (46.2%)	10 (38.5%)	2 (7.7%)	1 (3.8%)	0	0	1 (3.8%) *	
6.	Medical Humanities should continue to use team-based learning in future courses.							7.0
	0	0	0	1 (3.8%) *	2 (7.7%)	8 (30.8%)	15 (57.7%)	
7.	Writing reflections on our service-learning project fostered my professional development.							6.0
	1 (3.8%) *	0	0	0	6 (23.1%)	10 (38.5%)	9 (34.6%)	
8.	Encounters with people in our service-learning project caused me to study for all of my courses with more interest than likely would have occurred without the project.							4.0
	3 (11.5%)	0	4 (15.4%)	9 (34.6%)	4 (15.4%)	4 (15.4%)	2 (7.7%)	
9.	Encounters with people in our service-learning project will help me to be engaged with people regardless of the setting or disposition of the person.							6.0
	1 (3.8%) *	0	0	0	6 (23.1%)	8 (30.8%)	11 (42.3%)	
10.	Encounters with people/venues in our service-learning project helped me to see my potential biases toward people/venues more clearly.							6.0
	1 (3.8%) *	0	0	0	6 (23.1%)	10 (38.5%)	9 (34.6%)	
11.	Of what biases did you become aware during encounters with people/venues in your service-learning project?							

* Not including statistically significant outliers for items 1, 2, 3, 5, 7, 9, 10 (*p* < 0.01) and 6 (*p* < 0.05).

**Table 2 ijerph-16-03926-t002:** Summary of Studies.

Hypothesis	1. Service-learning experiences will be associated with dissonance, self-examination, bias mitigation, dissonance reconciliation, and compassionate behavior.	2. Students will express positive attitudes toward their teams, community service, and recognizing/mitigating their implicit biases in association with team service-learning experiences.	3. Reflections on team service-learning experiences will be accompanied by increases in students’ reflective capacity.	4. Students’ reflective capacity scores will correlate positively with their cognitive empathy scores (a component of compassion).
Cohort	One	One and two	Two	Three
Time of study	Aug–Dec 2017	Aug–Dec 2017/18	Aug–Dec 2018	Aug 2019
Method of data collection	Analysis of written reflections	Survey of attitudes	Reflective practice questionnaire (RPQ)	RPQ and Jefferson Scale of Empathy
Independent variable	Reflection on service-learning	Team reflection on service-learning	Reflection on service-learning	Reflective capacity scores
Dependent variable(s)	Reported dissonance, self-examination, bias mitigation, dissonance reconciliation, and compassionate behavior	Reported attitudes toward their teams, community service, and recognizing/mitigating their implicit biases	Reflective capacity scores	Cognitive empathy scores
Statistical analysis	Kruskal–Wallis	Kruskal–Wallis	ANOVA, *t*-test	Pearson *r* values

**Table 3 ijerph-16-03926-t003:** Summary of biases expressed by students (Item 11 of Table 1) in the survey about their team service-learning experiences (19 of 26 students stated one or more of their biases).

Nature of Negative Bias	Number of Times Expressed
Homeless People	5
Culture/Race	5
Older People	3
Attitudes of Other People	3
Gender	2
People with Serious Diseases	2
Other Healthcare Providers	2
Nonclinical Settings for Service	1
Hospital Volunteering	1
Socioeconomic Status	1

**Table 4 ijerph-16-03926-t004:** Pearson correlations among the Reflective Practice Questionnaire (RPQ) subscales for the MSBS students in August 2018.

	RC	DfI	CG	CC	Unc	SiP	JS
RC	1						
DfI	0.13	1					
CG	0.28	−0.14	1				
CC	0.62 ***	−0.13	0.58 **	1			
Unc	−0.33	0.23	−0.51 **	−0.62 ***	1		
SiP	−0.27	0.01	−0.32	−0.42 *	0.68 ***	1	
JS	0.57 **	0.31	0.24	0.51 **	−0.55 **	−0.48 *	1

* *p* < 0.05, ** *p* < 0.01, *** *p* < 0.001. Sub-scales: RC = reflective capacity; DfI = desire for improvement; CG = confidence—general; CC = confidence—communication; Unc = uncertainty; SiP = stress interacting with patients/clients; JS = job satisfaction.

**Table 5 ijerph-16-03926-t005:** Pearson correlations among the RPQ subscales for the MSBS students in December 2018.

	RC	DfI	CG	CC	Unc	SiP	JS
RC	1						
DfI	0.17	1					
CG	0.42 *	−0.29	1				
CC	0.70 ***	−0.08	0.43 *	1			
Unc	0.22	0.55 **	−0.04	0.00	1		
SiP	0.29	0.39 *	−0.07	0.12	0.84 ***	1	
JS	0.20	−0.09	0.06	0.17	−0.52 **	−0.50 **	1

* *p* < 0.05, ** *p* < 0.01, *** *p* < 0.001. Sub-scales: RC = reflective capacity; DfI = desire for improvement; CG = confidence—general; CC = confidence—communication; Unc = uncertainty; SiP = stress interacting with patients/clients; JS = job satisfaction.

**Table 6 ijerph-16-03926-t006:** Summary of findings/conclusions.

Hypothesis	1. Team service-learning experiences will be associated with dissonance, self-examination, bias mitigation, dissonance reconciliation, and compassionate behavior.	2. Students will express positive attitudes toward their teams, community service, and recognizing/mitigating their implicit biases in association with team service-learning experiences.	3. Reflections on team service-learning experiences will be accompanied by increases in students’ reflective capacity.	4. Students’ reflective capacity (RC) scores will correlate positively with their cognitive empathy scores (a component of compassion).
Cohort	One	One and two	Two	Three
Conclusion 1	All but one of 26 students reported dissonance, self-examination, bias mitigation, dissonance reconciliation, and compassionate behavior.	Virtually all 52 students overwhelmingly expressed positive attitudes toward their team.	Students’ mean reflective capacity, self-appraisal, and reflection-on-action scores increased significantly.	Students’ RC, self-appraisal, and reflection-on-action scores correlated strongly with their cognitive empathy scores.
Conclusion 2	Teams differed somewhat in their frequency of self-examination and bias mitigation.	Virtually all students mitigated their biases and reported professional development.	Students refocused their uncertainty and stress from negative associations with confidence to positive associations with desire for improvement.	
Conclusion 3		The vast majority of students agreed that their service-learning project was very engaging.	Reflective capacity went from negative to positive associations with uncertainty and stress.

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
