# Peer review of "Selecting and Performing Service-Learning in a Team-Based Learning Format Fosters Dissonance, Reflective Capacity, Self-Examination, Bias Mitigation, and Compassionate Behavior in Prospective Medical Students"

_ijerph, 2019, doi:10.3390/ijerph16203926_

Round 1
Reviewer 1 Report
This study is very important as it illustrates how to help improve self awareness of biases and address them. This is particularly important for (but not limited to) the health sector where future professionals need to not only have the right skills but also the right attitude. The study referred to several dimensions with a view to demonstrating the effectiveness of team based learning in helping induce attitudinal transformations and associated elements.
However there are a number of areas for consideration. Initially perhaps the authors could have made some references to views on the value of reflection (for example in educational setting) as well as philosophy of ethics and perhaps the positioning of this study within it (e.g. virtue ethics?).
There are issues that the authors need to be more upfront about. This may be prompted by the question: "were there some contributing factors to the results that were not looked at more closely?". Such factors may include 'internal' dimensions (such as 'personality' or past experience, as illustrated in appendix A statements) or 'current' environmental factors such as (teachers') expectations and ethos of the field of studies which may 'induce' students to be more willing to change. There is also the issue of 'transformation'. Cognitive psychologists tell us that there are certain dimensions that are due to core beliefs (deeply ingrained views of the world) which usually may take much longer to change. So how reliable are the claimed transformations?
There were also some issues of presentation clarity. For example the author should clarify the conceptualisation in a diagram with dependent and independent variables. This also links to the clarity of the design of the methodology that could have been helped by providing a representation of the data gathering steps (again, in a table or diagram format), showing the variables looked at each step for each cohort. In the concluding part, a table showing a summary of the findings could have been useful too.
Nevertheless we think this is a very good piece of study that many people working in the social/health sciences could benefit from.
Reviewer 2 Report
The study is interesting, current, and provides insights into areas in which there are clear deficits as far as the abilities of medical staff are concerned. However, the rationale of the study should be clearly described; additionally, conclusions presented in a broader context would be beneficial - to better fit into the profile of the journal. A strong point of the study is the mixed methodology used (quantitative and qualitative).
As for the issues (methodology):
the authors first quote "medical students" as participants, then, in the description of the study group, they relate to the "Master of Science in Biomedical Sciences" course - the rationale of the study is different if the studied subjects are future physicians or future technical/scientific staff not engaged in care for other people statistics: what are "nonparametric multiple comparison tests"?Results:
statistics: the authors do not justify why they select medians as main outcome measures Table 2: "elderly" is currently considered stigmatizing; consider using "older" insteadAuthor Response
Please see attachment.
